

# Extracellular enzyme production in the coastal upwelling system off Peru during different upwelling scenarios: a mesocosm experiment

Kristian Spilling[1,2,*], Jonna Piiparinen[1], Eric P. Achterberg[3], Javier Arístegui[4], Lennart T. Bach[5], Maria T. Camarena-Gómez[1], Elisabeth von der Esch[6], Martin A. Fischer[7], Markel Gómez-Letona[4], Nauzet Hernández-Hernández[4], Judith Meyer[3], Ruth A. Schmitz[7], Ulf Riebesell[3]

1. Marine Research Centre, Finnish Environment Institute, Helsinki, Finland

2. Centre for Coastal Research, University of Agder, Kristiansand Norway

3. GEOMAR Helmholtz Centre for Ocean Research Kiel, Kiel, Germany

4. Instituto de Oceanografía y Cambio Global, IOCAG, Universidad de Las Palmas de Gran Canaria, Las Palmas de Gran Canaria, Spain

5. Institute for Marine and Antarctic Studies, University of Tasmania, Tasmania, Australia

6. Institute of Hydrochemistry, Chair of Analytical Chemistry and Water Chemistry, Technical University of Munich, Munich, Germany

7. Institute for General Microbiology, Christian Albrechts University Kiel, Germany

*corresponding author: kristian.spilling@syke.fi





Abstract
The Peruvian upwelling system is a highly productive ecosystem that could be altered by
ongoing global changes. We carried out a mesocosm experiment off Peru, with the addition of
water masses from the regional oxygen minimum zone (OMZ) collected at two different sites
simulating two different upwelling scenarios. Here we focus on pelagic remineralization of
organic matter by extracellular enzyme production of leucine aminopeptidase (LAP) and alkaline
phosphatase activity (APA). After addition of the OMZ water, dissolved inorganic nitrogen (N)
was depleted, but the standing stock of phytoplankton was relatively high even after nutrient
depletion (mostly >4 µg chlorophyll $a$ L$^{-1}$). During the initial phase of the experiment, APA was
0.6 nmol L$^{-1}$ h$^{-1}$ even though the PO$_4^{3-}$ concentration was >0.5 µmol L$^{-1}$. Initially, the dissolved
organic phosphorus (DOP) decreased, coinciding with an increase in PO$_4^{3-}$ concentration
probably linked to the APA. The LAP activity was very high with most of the measurements in
the range 200-800 nmol L$^{-1}$ h$^{-1}$. This enzyme degrades amino acids, and these high values are
probably linked to the highly productive, but N-limited coastal ecosystem. Also, the experiment
took place during a rare coastal El Niño event with higher-than-normal surface temperatures,
which could have affected the enzyme production. Using a non-parametric multidimensional
scaling analysis (NMDS) with a generalized additive model (GAM), we found that
biogeochemical variables (e.g. nutrient and chlorophyll $a$ concentrations), phytoplankton and
bacterial communities explained up to 64% of the variability in APA. The bacterial community
explained best the variability (34%) in LAP. The high hydrolysis rates for this enzyme suggests
that pelagic N remineralization supported the high standing stock of primary producers in the
mesocosms after N depletion.



Introduction

The Peruvian upwelling system is one of the most productive marine ecosystems in the world
(FAO, 2018). Its high productivity is driven by the upwelling of deep, nutrient rich water that
fuels primary production when reaching the sunlit surface ocean. The fate of the biomass
produced is of great importance for higher trophic levels and biogeochemical cycles. The
primary limiting nutrient is nitrogen (N), but iron (Fe) availability is also an important driver for
phytoplankton biomass production in addition to light (Chavez et al., 2008; Messié and Chavez,
2015). Part of the phytoplankton biomass passes to higher trophic levels through grazing and
predation. As the upwelled water parcel is transported further offshore by Ekman transport, part
of the biomass settles out of the euphotic zone and is decomposed in intermediate water layers
creating an extensive oxygen minimum zone (OMZ; Kalvelage et al., 2013).
The ongoing warming of surface waters is projected to have several consequences on marine
ecosystems. For example, increasing temperatures lead to a reduction in gas solubility causing a
decrease in oxygen concentrations; warming will also increase thermal stratification and reduce
the ventilation of the deeper ocean (Keeling et al., 2010). Both of these effects will lead to
expanding OMZs with potential consequences for biogeochemical cycling (Oschlies et al.,
2018). Biogeochemical cycles of nitrogen (N) and phosphorus (P) are affected by $O_2$ depletions,
e.g. through denitrification and sediment P release (Canfield et al., 2005). Hence, expanding
OMZs may decrease the inorganic N : P ratio in the upwelled water potentially affecting the
seston (i.e. all suspended particles) stoichiometry and plankton community composition (Hauss
et al., 2012; Spilling et al., 2019).



After inorganic nutrients (primarily N) have been depleted, the productive surface layer is driven
by recycled production. In this process, dissolved organic matter (DOM) must first be broken
down into simpler forms before the DOM elements become biologically available. The
decomposition of DOM is not a uniform process as it is affected by both abiotic and biotic
variables. Extracellular enzymes hydrolyze complex dissolved organic molecules and is the first
step in remineralization of these DOM elements (Arnosti, 2011). Quantifying the rates of pelagic
remineralization is important for understanding recycled production and element fluxes in the
uppermost water masses. There are a range of different enzymes that are used for hydrolyzing
DOM, and two of the most studied ones are Leucine aminopeptidase (LAP) and Alkaline
phosphatase (AP).
LAP is a protein degrading enzyme that is used extracellularly in aquatic systems by bacteria,
some phytoplankton and fungi (Hoppe et al., 1988; Stoecker and Gustafson, 2003; Gutiérrez et
al., 2011). It hydrolyses a broad spectrum of substrates with a free amino group, but it has
preference for N-terminal leucine and related amino acids (Burley et al., 1990).
The AP enzyme is produced by a wide range of different organisms including aquatic bacteria
and phytoplankton. Its main function is related to the hydrolysis of phosphate monoesters that
separate orthophosphate ($PO_4$) from an organic compound (Perry, 1972; Hoppe, 2003). AP exists
either as ectoenzyme (on the cell wall) or is excreted extracellularly and has for phytoplankton
commonly been related to P-limitation in aquatic environments (Rose and Axler, 1997; Nausch,
1998). Bacterial AP activity (APA) is more complex, as some, in particular particle attached
bacteria, take up and use C and N from the organic molecule after hydrolysis, and may for this
reason produce AP even under P replete conditions (Benitez-Nelson and Buesseler, 1999;
Hoppe, 2003; Labry et al., 2016).





In this study, we were interested in the dynamics of LAP and APA after an upwelling event in
relation to biogeochemical variables and communities of plankton and bacterioplankton, and our
main aim was to understand how much of the variability in enzyme activities could be explained
by biogeochemical variables (e.g. nutrient concentrations) and microbial communities. This was
done during a mesocosm experiment set up off the coast of Peru.

Materials and methods
A detailed description of the mesocosm set up and collection and addition of deep-water can be
found in Bach et al. (2020) within this special issue. In short, the mesocosm bags were 2 m in
diameter and extended from the surface down to 19 m depth, where the last 2 m was a conical
sediment trap. Eight mesocosm bags were used and they were moored at 12.0555°S; 77.2348°W
just north of Isla San Lorenzo where the water depth is ~30 m. The mesocosms were closed by
attaching the sediment trap to the bottom and pulling the top above the surface on 25 Feb, 2017.
The bags were regularly cleaned from the inside and outside and sampled every second day with
integrated water samplers (0-10 m depth, IWS, Hydro-Bios). For a full detailed sampling and
cleaning timetable see Bach et al. (2020).
Water (100 m$^3$) from the oxygen minimum zone (OMZ) was collected from two locations. The
first was collected on day 5 from 12.028323°S; 77.223603°W from 30 m depth, and the second
one from 12.044333°S; 77.377583°W from 90 m depth. The original aim was to collect severe
and moderate OMZ signature water (differing in e.g. nitrate concentrations) from the first and
second site, respectively. This assumption was based on long-term monitoring data, however, the
chemical properties (e.g. nitrate concentration) was more similar in these water masses than





anticipated reflecting low and very low OMZ signatures from site 1 and 2 respectively, but this
was discovered only after the collection. Deep-water was added to the mesocosms in two steps
on day 11 and 12 after the enclosure of the mesocosms. Approximately ~20 m$^3$ of the mesocosm
water was exchanged with OMZ water, and both deep-water stations were pumped into four
replicate mesocosms. The water removed was pumped out from 11-12 m depth whereas the
deep-water was pumped into carefully moving the input hose between 14-17 m depth. The water
collected at 30 m depth was pumped into mesocosms M1, M4, M5 and M8 having low OMZ
signature and deep-water from 90 m depth into mesocosms M2, M3, M6 and M7 having very
low OMZ signature.
At the site of the mesocosms, the OMZ is close to the surface (<10 m depth; Graco et al., 2017)
and consequently the bottom part of the mesocosm was low in oxygen. In order to keep the
stratification inside the mesocosm we added 69 L of concentrated brine on day 13 by carefully
inserting it between 12.5-17 m depth. The same procedure was repeated on day 33 when 33 L of
brine was added. This artificial halocline prevented complete mixing of the mesocosm and the
lower part of the mesocosm had a very different water chemistry compared to the upper 10 m. At
the end of the experiment a third addition of brine was carried out to measure the total volume of
the mesocosms.

Nutrient concentrations
Inorganic nutrients were determined from filtered (0.45 µm filter, Sterivex, Merck) samples
immediately after the water arrived in the laboratory. For the measurements, we used a
continuous flow analyzer (QuAAtro AutoAnalyzer, SEAL Analytical) connected to a





fluorescence detector (FP-2020, JASCO). Phosphate ($PO_4^{3-}$), nitrate ($NO_3^-$) and nitrite ($NO_2^-$)
were determined colorimetrically (Murphy and Riley, 1962; Morris and Riley, 1963) and
corrected with the refractive index method reported by Coverly et al. (2012). Ammonium ($NH_4^+$)
concentrations were determined fluorometrically (Kérouel and Aminot, 1997). Dissolved
inorganic nitrogen (DIN) was calculated by summing $NO_3^-$, $NO_2^-$ and $NH_4^+$. Further details on
measurement accuracy can be found in Bach et al. (2020).
To measure total dissolved nitrogen (TDN) and phosphorus (TDP), the samples were first
filtered through pre-combusted (5 h, 450°C) Whatman GF/F filters (pore size 0.7 µm). The
filtrate was collected in 50 mL acid-cleaned high-density polyethylene (HDPE) bottles and
placed directly into a freezer (-20°C). Later the filtrates were thawed at room temperature over a
period of 24 hours and divided in two. The first half was used to determine inorganic nutrient
concentrations as described above. From the other half we determined the TDN and TDP
concentrations. An oxidizing reagent (Oxisolv, Merck) was added, and the samples were
autoclaved for 30 minutes. TDN and TDP were measured spectrophotometrically (QuAAtro,
Seal Analytical). Dissolved organic nitrogen (DON) concentrations were calculated by
subtracting DIN from TDN. Dissolved organic phosphorus (DOP) was calculated as the
difference between TDP and $PO_4^{3-}$.

Fluorescent dissolved organic matter and PARAFAC analysis
Fluorescent dissolved organic matter (FDOM) was determined by measuring fluorescence in
water samples with a Cary Eclipse (Agilent Technologies) spectrofluorometer, using excitation



and emission slit widths of 10 nm. Wavelength ranges were set to 230-456 nm for excitation,
with 2 nm increments, and the 290-600 nm for emission with 5 nm increments. The
measurements were collected into excitation-emission matrices (EEM). Blanks were measured
with the same settings using ultrapure water.
Raw measurements were processed using the DOMFluor toolbox (v. 1.7; Stedmon and Bro,
2008) for Matlab (R2017a). The processing consisted in 1) blank subtraction from seawater
EEMs, 2) EEMs normalization to the Raman area (RA), estimated applying the trapezoidal rule
of integration on the emission scan at the 350 nm excitation wavelength in the blank EEMs, and
3) cropping of the 1st and 2nd order Rayleigh scatter bands. Inner filter correction was not
performed as for the duration of the experiment the absorption coefficient at 250 nm (a250)
displayed values (mean ± sd = 1.56 ± 0.91 m$^{-1}$) well below 10 m$^{-1}$, above which correction is
considered necessary (Stedmon and Bro, 2008).
The processed EEMs were analyzed applying a Parallel Factor Analysis (PARAFAC) using the
DOMFluor toolbox. The PARAFAC model was constructed based on 125 samples (outliers were
removed) and validated using split-half validation and random initialization. The resulting model
consisted of 4 components (C1-C4; supplementary material Fig S1). For each of them, the
fluorescence maximum (Fmax) was recorded. The identified fluorophores were compared to
others found in the literature using the OpenFluor database (openfluor.lablicate.com; Murphy et
al., 2014).




Phytoplankton community and chlorophyll a
Flow Cytometry subsamples were transferred from the IWS into 50 mL beakers and stored cool
in the dark until analysis max. 8 hours after sampling. Each sample (650 µL) was analyzed with
an Accuri C6 flow cytometer (BD Biosciences) set to a high flow rate (i.e. 66 µL/min).
Phytoplankton groups were differentiated based on the strength of the forward scatter (FSC-A),
the side scatter (SSC-A), the red fluorescence (FL3-A) and orange fluorescence (FL2-A) signal (
"A" refers to the area of the signal integral). Furthermore, we used sequential filtrations with
different polycarbonate filters (Whatman, pore-sizes 0.2, 0.4, 0.8, 2, 3, 5, 8 µm) to distinguish
populations in the cytogram based on size. This procedure was helpful to approximate how FSC-
A values corresponded with size. We defined the following phytoplankton groups:
Synechococcus-like cells (Syn; 0.2-2µm), Cryptophyte-like cells (Crypto; ~90% between 2-5
µm), picoeukaryotes (Peuks; 0.2-2 µm), Nanoeukaryotes (Nano; 2-20 µm, mostly in the lower
range), Microeukaryotes 1 (Mikro1; ~15-40 µm, occasionally overlapping with Nano),
Microeukaryotes 2 (Mikro2; ~>40 µm, cluster dominated by *Akashiwo sanguineum* from about
day 20 onward), elongated cells "chains" determined by the ratio of FSC-A to FSC-H where "H"
refers to the height of the forward scatter signal (details about this approach are provided in Paul
et al., this issue. The goal of this was to detect chain-forming diatoms which we expected to be
an important component of the community).
Samples for chlorophyll *a* (chl-*a*) determination were filtered onto GF/F filters (Whatman) and
flash frozen in liquid nitrogen and stored at -80 °C (or dry ice for a brief period during air
transfer; ~2 days) until measurement. The chl-*a* concentration was measured using high-
performance liquid chromatography. The chl-*a* autofluorescence of the phytoplankton
community was measured with a handheld fluorometer (AquaPen, Photon Systems Instruments)



using 450 nm excitation light. The photochemical efficiency was calculated based on the
relationship between the variable to maximal fluorescence (Fv/Fm).

16S-rRNA gene based bacterial community determination
One liter of surface water obtained from the individual sampling sites was filtered through sterile
Millipore Express PLUS membrane filters (polyethersulfon) with a cut-off of 0.22 µm and a
diameter of 47 mm (Merck Millipore). After filtration, the filters were flash frozen in liquid
nitrogen and stored at -80°C until nucleic acid extraction. Nucleic acid extraction was performed
using the NucleoSpin TriPrep- Kit (Machery-Nagle) according to manufacturer's instruction
with an additional step at the beginning of the extraction using a pestle to properly homogenize
the sample.
Primers applied for the amplification of the bacterial 16S rRNA gene fragments were annealing
to the variable region 1 and 2 and consisted of an initial standardized Illumina adapter (regular),
followed by an 8 nucleotide barcode (X's), a linker region (underlined) and a primer sequence
(bold). The sequences were for the forward primer Bac27 5'-
AATGATACGGCGACCACCGAGATCTACACXXXXXXXXTATGGTAATTGT**AGAGTTT**
**GATCCTGGCTCAG**-3' and reverse Bac338 5'-
CAAGCAGAAGACGGCATACGAGATXXXXXXXXAGTCAGTCAGCC**TGCTGCCTCCC**
**GTAGGAGT**-3'. The individual PCR reaction contained 100 ng of the extracted DNA. PCR
conditions and purification of the amplification product were previously described (Fischer et al.
2019a). The final library pool for sequencing was combined from the eluates and contained 100
ng of DNA. Amplicon library sequencing was performed on a MiSeq instrument. Library



therefore was prepared according to the manufacturer's instructions and sequenced using the v3
chemistry with 2 x 300bp paired-end.
Reads generated with amplicon sequencing were trimmed using the trimmomatic software
version 0.33 (Bolger et al., 2014) as described in Fischer et al. (2019b). Briefly, reads were
analyzed with a sliding window of 4 bp and regions were trimmed if the average Phred score
(Ewing and Green, 1998; Ewing et al., 1998) within the window was below 30. Trimmed reads
were kept within the dataset if the forward and reverse read both survived the quality trimming
and were longer than 36 bp. Afterwards, 20,000 reads per sample were kept in the dataset
(exceptions were sample M1 on day 10 (5817 reads) and M7 on day 24 (17660 reads) for further
analysis.
Quality trimmed sequences were analyzed using MOTHUR software, version 1.35.1 (Schloss et
al., 2009) as described in Fischer et al. (2019a). The quality filtered and subsampled reads were
concatenated to 1,040,321 contiguous sequences (contigs) using the command make.contig.
Contigs were filtered for ambiguous bases, homopolymers longer than 8 bases or sequences
longer than 552 bases using the command screen.seqs. The resulting 754,310 contigs were
checked for redundant sequences using the command unique.seqs and clustered to 199,746
unique sequences. The sequences were consecutively aligned to a modified version of the
SILVA database release version 132 (Pruesse et al., 2012) containing only the hypervariable
regions V1 and V2 by the command align.seqs. Sequences not aligning in the expected region
were removed from the dataset using the command screen.seqs. The alignment was further
optimized by removing gap-only columns with the command filter.seqs. The alignment
contained 717,217 sequences (148,760 unique). Rare and closely related sequences were
clustered using the commands unique.seqs and precluster.seqs. The latter was used to cluster



sequences with up to 3 positional differences compared to larger sequence clusters together.
Chimeric sequences were removed using the implemented software UCHIME (Edgar et al.,
2011) using the command chimera.uchime, followed by remove.seqs leaving 551,142 sequences
(29,519 unique) in the dataset. The classification of the sequences was performed against the
SILVA database and was done with a bootstrap threshold of 80 %. Operational taxonomic units
(OTUs) were formed using the average neighbor clustering method with the command
cluster.split. A sample-by-OUT table on the 97 % level, containing 10,258 OTUs, was generated
using the command make.shared. These OTUs were used for the subsequent analysis. After the
removal of mitochondria, chloroplast and singletons, 3225 OTUs were retained. These OTUs
were used for downstream analysis.

Extracellular enzymes
The leucine aminopeptidase (LAP) activity was determined using the method described by
Stoecker and Gustafson (2003) using *L*-leucine 7-amido-4-methyl-coumarin (Leu-AMC; Sigma
Aldrich) as a substrate. Leu-AMC was added to a final concentration of 500 µmol $L^{-1}$, which was
determined in separate kinetics tests to saturate the enzyme activity. The samples (100-200 µl)
were incubated in the dark at in situ surface temperature for a minimum of four hours. The
fluorescence was measured every 30-60 min with a Cary Eclipse (Agilent Technologies)
spectrofluorometer using 380 nm excitation and 440 nm emission wavelengths. The results were
compared with a standard curve determined using 7-amino-4-methyl-coumarin (AMC; Sigma
Aldrich) dissolved in DMSO, and the LAP activity calculated by linear regression.





Measurements of alkaline phosphatase activity (APA) were conducted with 20 ml subsamples of
initial/incubated seawater using 100 nmol $L^{-1}$ 4-methylumbelliferyl phosphate (MUF-P; Sigma-
Aldrich) as the organic phosphate substrate (Ammerman, 1993). Fluorescence was measured on
a BIOTEK Microplate Reader with a Cary Eclipse (Agilent Technologies) spectrofluorometer
using 355 nm excitation light and 460 nm emission detection. Following MUF-P addition,
fluorescence measurements were performed at 0, 1.5, and 3 h and APA ($h^{-1}$) was calculated
from the linear increase in fluorescence and calibrated against 4-methylumbelliferone (MUF;
Sigma-Aldrich). The assays were performed and incubated in the dark. Ultrapure water (Milli-Q)
blanks and paraformaldehyde-killed controls generally yielded fluorescence values similar to $t =$
0 readings.

Statistical analysis
Before comparisons between the two experimental treatments were conducted, we first
constructed a cumulative value where each measured value was summed up for each sampling
day. The linear regressions of the cumulative enzyme activity from the two treatments (n = 4)
were compared with Student's t-test. In addition, the effect of biogeochemical, phytoplankton
and bacterioplankton community composition to APA and LAP was determined, using the
ordination scores of the first and second axis of a non-parametric multidimensional scaling
(NMDS) as explanatory variables in generalized additive models (GAMs) with APA or LAP as
dependent variable. The NMDS was applied separately to each group of variables:
biogeochemical, phytoplankton community and bacterioplankton community. The individual



explanatory power of each MDS score was estimated with a univariate GAM. The visualization
of the links was done for each explanatory variable through the prediction from the full model
object, setting all other explanatory variables at their mean value. In addition, links to the scores
of the biogeochemical variables and phytoplankton community NMDS were estimated with one
GAM model. It was not possible to include the bacterioplankton community into this model due
to the different sampling regime (lower number of samples) and this was treated with a second
model. NMDS was estimated with the metaMDS function in the Vegan package (Oksanen et al.,
2019), and GAMs were fitted using the gam function in the mgcv package (Wood, 2017). For
explaining the deviance, an adjusted coefficient of determination ($R^2$) was used. An adjusted $R^2$
takes into account the model complexity and is more conservative than a non-adjusted $R^2$.

RESULTS

Nutrients
The addition of deep-water increased the phosphate concentrations whereas the dissolved
inorganic nitrogen (DIN) was >2 µmol L$^{-1}$ in the mesocosms until after the addition of deep-
water (days 11 and 12 of the experiment). After the addition of the deep-water, the DIN
concentration rapidly declined and was depleted at day 15 in most mesocosms except in M3
where DIN depletion occurred a week later (day 22; Fig 1). The PO$_4^{3-}$ concentration increased
after closing the mesocosm and reached ~1.9 µmol L$^{-1}$ in all mesocosms after the deep-water





addition. There was only a slight reduction to approximately 1.5 µmol $PO_4^{3-}$ $L^{-1}$ over the course
of the experiment (Fig 1).
The dissolved organic nitrogen (DON) and phosphorus (DOP) concentrations were initially 9 –
12 µmol $L^{-1}$ and 0.6 – 1.0 µmol $L^{-1}$, respectively. There was no drastic change in DON with the
deep-water addition and there was an overall decrease in DON to 6.0 - 7.9 µmol $L^{-1}$ on day 30
after which it increased somewhat again. The DOP concentrations decreased rapidly the first 8
days to 0.19 - 0.32 µmol $L^{-1}$ but increased after the deep-water addition and remained within 0.2
- 0.7 µmol $L^{-1}$ interval for the rest of the experiment.
The PARAFAC modelling of the EEMs yielded four FDOM components (C1-C4; Fig 2, Fig S1).
Using the OpenFluor database we identified multiple fluorophores with strong similarity
(TCCex·em > 0.95) to our components (Table S1). Components 1 and 3 had characteristics
resembling amino acid/protein compounds whereas components 2 and 4 were more humic-like
(Table S1). All FDOM components increased sharply at day 18. This did not take place in
Pacific seawater sampled outside the mesocosm where the FDOM was relatively stable
throughout the experiment. After the increase at day 18, humic-like components (C2 and C4)
were relatively stable but decreased slightly after day 28-30. The amino acid-like components
(C1 and C3) exhibited higher variability among mesocosms, and C3 had overall higher
variability throughout the experiment. Both humic-like and amino acid-like components
maintained fluorescence values above the initial ones until the end of the experiment, but there
were no clear differences between the treatments. However, towards the end of the experiment
M1 and M2 had highest concentrations of C1, M1 also had highest concentration of C2 and C3
whereas M3 had the highest concentration of C4 at the end of the experiment.




Chlorophyll, photochemical efficiency and phytoplankton community

After deep-water addition, the chl-*a* concentration increased from 2-4 µg L$^{-1}$ to 4-8 µg L$^{-1}$ except

for mesocosms M3 and M4 where the increase was not as pronounced (Fig 3). The chl-*a*

concentration in M3 increased after day 22 to ~4 µg Chl-*a* L$^{-1}$, whereas in M4 the chl-*a*

concentration remained low (<2 µg L$^{-1}$) throughout most of the experiment (Fig 3). The

photochemical efficiency (Fv/Fm) was approximately 0.7 throughout the whole experiment

without major difference between mesocosms, except for M4 where it was consistently lower

(<0.6) during the last week of experiment (Fig 3).

The initial community was dominated by diatoms in terms of biomass but this group gradually

reduced in numbers after the enclosure of the mesocosms and instead the mixotrophic

dinoflagellate *Akashiwo sanguineum* appeared (Fig 4). The cell counts done with the flow

cytometer were checked with a microscope and this was the primary species in terms of biomass

in the Microeukaryote 2 group (Fig 4). The exceptions were mesocosms M3 and M4 where this

dinoflagellate was not abundant (M4) or bloomed later (M3) and where there were more

Chrysophytes. In M4 there was in addition a bloom of picoeukaryotes starting after day 20 (Fig

4).

Bacterial community

The bacterial community was dominated by the class Alphaproteobacteria throughout the whole

experiment and in all the mesocosms units, reaching values between 60 to 88% of the total



sequences at day16 (Fig 5). Within Alphaproteobacteria, the *Roseobacter* lineage (genera
HIMB11, *Ascidiaceihabitans*, *Amylibacter* and *Planktomarina* in M1) of the order
Rhodobacterales contributed most to the bacterial community in all the mesocosms (10-55 %) in
particular on day 16, except in M8 where the SAR11 Ia clade dominated the community (55% of
the total sequence at day 16). The order Parvibaculales had high relative abundances (12-20% of
the total sequences) in M4, M5, M6 and M7 before the deep-water addition (day 10) decreasing
in the following week. The relative abundance of order Rickettsiales peaked at day 16 in all the
mesocosms except in M8, decreasing after one week. The class Gammaproteobacteria comprised
between 20 to 45% of the total relative abundance. Within Gammaproteobacteria, the order
Thiomicospirales had high relative abundance (8-17% total sequences) at day 10 in most of the
mesocosms, whereas the order Cellvibrionales and order Oceanospirillales (genus
*Pseudohongiella*) increased from day 24 and by the end of the experiment, respectively. In M8,
the abundances of orders Thiomicospirales and Pseudomonadales (14% of total sequences)
increased at day 24. Other groups that increased in abundance in the second half of the
experiment were the deltaproteobacterial orders Desulfobacteriales (7-20% in M2, M3, M4 and
M5) and Bdellovibrionales (5-8% in M2, M3 and M4). The order Flavobacteriales dominated
within Bacteroidetes and the relative abundance ranged from 1 to 25% throughout the
experiment, being generally high (10-20%) at day 10 . The flavobacterial genus *Aurantivirga*
contributed > 7% in M1, M2 and M3.

Enzyme activity





The initial LAP activity before the deep-water addition was relatively low (average 359 nmol $L^{-1}$
$h^{-1}$ ± 81 nmol $L^{-1}$ $h^{-1}$ SD) but increased after the addition of deep-water in some of the
mesocosms (Fig 6). In M3 the LAP activity was high, reaching 1600 nmol $L^{-1}$ $h^{-1}$ directly after
the deep-water addition, but decreased after that. The highest cumulative LAP activity at the end
of the experiment was in M7 where the LAP activity was 716 nmol $L^{-1}$ $h^{-1}$ after deep-water
addition and the average after day 16 was 657 nmol $L^{-1}$ $h^{-1}$ ± 142 nmol $L^{-1}$ $h^{-1}$ (SD). There was a
difference between the treatments in the cumulative LAP after the addition of the deep-water
until day 16, with the very low OMZ signature (lowest $NO_3$ concentration) water producing the
highest LAP activity (Students' t-test, $p = 0.047$), but this difference disappeared after day 16 ($p$
$= 0.44$).
The alkaline phosphatase activity (APA) was 0.5-0.6 nmol $L^{-1}$ $h^{-1}$ at the beginning of the
experiment but decreased to undetectable levels after day 30 (Fig 7). There was a noticeable drop
in APA after the addition of the deep-water, and the decrease continued gradually until day 28
after which the APA was very low (<0.1 nmol $L^{-1}$ $d^{-1}$). The APA was similar in all the
mesocosms and there was no treatment effect ($p = 0.81$). The exception to this was M3 where the
APA was lower, compared to all other mesocosms for most of the experiment (Fig 7).
The variability in APA was better explained by the measured variables than LAP (Fig 8). The
biogeochemical variables and bacterioplankton community separately explained 62% of the
variability in APA, whereas the phytoplankton community alone explained 57% of the
variability. Combining both the biogeochemical variables and the phytoplankton community
increased the explanatory power to 74% (bacterioplankton community not included as the
number of sample points were less). The variability in LAP was best explained by the
bacterioplankton community (38%) followed by biogeochemical variables (20%) and



phytoplankton community (18%). The combined biochemical variables and phytoplankton
community explained 28% of the LAP variability.


DISCUSSION
After the closure and addition of deep-water there was rapid phytoplankton growth in the upper 5
m of the mesocosms, with low light conditions limiting primary production deeper down (Bach
et al., 2020). The DIN concentrations were depleted around day 18 coinciding with an increase in
several of the FDOM components (both amino acid-like and humic-like components), also
matching the end of the phytoplankton bloom. There was, however, relatively constant and low
export of carbon out from the mesocosms (Bach et al., 2020) and at the same time relatively high
Chl-*a* concentration (mostly >4 µg chl-*a* L$^{-1}$) under conditions with depleted DIN (Fig 3). In
addition, the photochemical efficiency was overall relatively high (>0.5) throughout the
experiment suggesting regenerated primary production driven by recycling of nutrients. The
measured hydrolysis rates, particularly LAP, indicated that extracellular enzyme activity plays an
important role for this recycled production.
The main aim of this study was to relate the biogeochemical and microbial community to the
extracellular enzyme activity and a more detailed description of the temporal development and
biomass comparison of microbial groups will be presented elsewhere in this special issue (e.g.
Bach et al., 2020). Among phytoplankton, diatoms are typically dominating following upwelling
events (Anabalón et al., 2016), whereas dinoflagellates tend to become more dominant after





establishment of stratification (Margalef et al., 1979). This was also seen in our mesocosm as the
dinoflagellate *Akashiwo sanguinea*, a mixotrophic species that may form red tides (Jeong et al.,
2005; Badylak et al., 2014), that quickly appeared in most mesocosm after OMZ water was
added with some exceptions. In M3 it appeared a little later and in M4 it did not bloom at all.
Interestingly these two mesocosms had a higher concentration of cryophytes and M4 had
additionally a bloom event of picoeukaryotes. Being mixotrophic, *A. sanguinea* is known to prey
on smaller species (Jeong et al., 2005) and lower grazing pressure could be the reason for the
bloom of picoeukaryotes in M4.
The bacterial community composition changed during the experiment but without clear treatment
effects. The dominant bacterial groups were the class Alphaproteobacteria, (Parvibaculales,
SAR11 subclade Ia, Roseobacter clade and Rickettsiales), class Gammaproteobacteria (SAR116
clade, Cellvibrionales, Oceanospirillales and SUP05 clade) and to lesser extent the class
Deltaproteobacteria (Desulfobacterales) and class Bacteroidea (order Flavobacteriales). SAR11
subclade Ia, Roseobacter clade, SAR116 clade, SUP05 clade and Desulfobacterales are known to
utilize inorganic and organic sulfur components such as hydrogen sulfide ($H_2S$), sulfate ($SO_4$)
and dimethylsulfoniopropionate (DMSP) metabolites for their metabolic requirements (Nowinski
et al., 2019) and are coupled with the nitrogen cycle (Schunck et al., 2013). Specifically, the
sulfur-oxidizing SPU05 oxidizes $H_2S$ coupled with the nitrate reduction and potentially produces
nitrite (Shah et al., 2017), whereas Desulfobacterales play an important role in $N_2$ fixation (Gier
et al., 2016). These bacterial taxa associated with the sulfur cycle are typically found in the OMZ
regions (Pajares et al., 2020). We observed a temporal shift in the bacterial community through
the experiment changing between sulfur-oxidizing (SUP05) and sulfate-reducing
(Desulfobacterales) bacteria, probably liked to the nitrate availability, i.e. more DIN at the





enclose of the mesocosms and thus more relative abundance of SUP05. We also observed a shift
within phytoplankton-associated bacteria (*Roseobacter* lineage, Gammaproteobacteria, and
Flavobacteriales) that likely responded to the availability of DOM supply during the experiment
(Buchan et al 2014, Chafee et al 2017). The high relative abundance of Flavobacteriales and
genera from the *Roseobacter* lineage on days 10 and 16, respectively, coincided with the increase
in chl-*a* and high LAP activity until day 16. Positive correlations have been observed between
chl-*a*, Bacteroides and Deltaproteobacteria and LAP during phytoplankton blooms (Shi et al
2019). However, we do not have gene expression data and cannot make any firm conclusion
about the connection between these groups and production of LAP.
The temporal shift in the bacterial community indicates niche partitioning between bacterial taxa
that assimilate different organic substrates or inorganic sulfur components, produced during
phytoplankton bloom events or from sulfidic events (Schunck et al., 2013; Callbeck et al., 2018;
Nowinski et al., 2019). Our results support previous studies that have demonstrated the important
role of the sulfur cycle in shaping the bacterial community composition in poorly oxygenated
waters (Schunck et al., 2013; Aldunate et al., 2018). It is worth to note that the conditions in the
bottom of the mesocosms were sub-oxic and there might have been a clear depth gradient in the
bacterial community that was not picked up by our integrated 0-10 m sampling.
Overall, there was a treatment effect of the different OMZ waters on the LAP activity, with
higher LAP in the very low OMZ signature addition, but this effect was only observed right after
the addition of the deep-water. There were also slightly higher $NO_3$ concentrations in this water
(Bach et al., 2020). However, this difference in both DIN and LAP was relatively small and
disappeared a week after the OMZ water addition, most likely because the collected deep-water
were more similar between the two locations than anticipated, with relatively similar





concentrations of DIN. Although there were differences between individual mesocosms in terms
of the plankton community structure, there were no clear differences between treatments, and we
can conclude that the availability of nutrients by itself can shift the LAP production.
The LAP activity in our study was very high (~10-times higher compared with most literature
data). In a comparable study but further offshore in Peru, the LAP activity was $20 – 65$ nmol L$^{-1}$
h$^{-1}$ in surface waters (Maßmig et al., 2020). Further to the south, in Chile (30° 30.80' S), values
up to 230 nmol L$^{-1}$ h$^{-1}$ have been recorded, with a clear seasonal cycle linked to upwelling events
(Gutiérrez et al., 2011). With most of our data ranging between $200 – 800$ nmol L$^{-1}$ h$^{-1}$ it is clear
that these LAP activities are linked to the upwelling, which is more intense near the coast and
also more constant at the study site compared with further south. The enzyme activity in
sediments can be up to three orders of magnitude higher than what we found (Hoppe et al.,
2002), and an order of magnitude higher values have been observed in a eutrophic, salt-water
lake (Song et al., 2019). The high LAP activities are likely a reflection of the high microbial
activity in the Peruvian upwelling system. The experiment was also taking place during a rare
coastal El Niño event (Garreaud, 2018), with anomalous higher surface temperatures, which
could be a reason for the high values we recorded as LAP is known to increase with temperature
(Christian and Karl, 1995).
There was also some loss of N due to denitrification, estimated to 0.2-4.2 nmol N$_2$ L$^{-1}$ h$^{-1}$ during
the experiment (Schulz et al 2021). For comparison, the LAP activity suggests an average of 417
nmol L$^{-1}$ h$^{-1}$ hydrolyzation of N-containing compounds, but this should be seen as the maximal
potential rather than the actual rate. The use of fluorescently labelled substrates for measuring
extracellular activity is a proxy method that has some drawbacks. The primary one is that the
molecular structure of the substrate used is never equivalent to the high molecular weight DOM



in the water. This means that the measured hydrolysis rates could be an overestimation of the
actual hydrolysis rates of DOM (e.g. Arnosti, 2011). The primary benefit of the method is that it
is straightforward and has been in widespread use for decades, which means that comparisons
with other ecosystems is possible, and for our purpose, we can use it for better understanding
how much of the variability can be explained by the other measured variables.

Considering the APA, the most interesting aspect was that it was measurable in the beginning of
the experiment at high $PO_4^{3-}$ concentration. This high APA activity at high $PO_4^{3-}$ concentration
has been observed in deep oceans (Hoppe and Ullrich, 1999; Baltar et al 2016). Baltar and
collaborators (2016) also observed an increase in APA in experiments amended with organic
matter suggesting the activity of APA by organic matter supply, independently of the $PO_4^{3-}$
concentration. This could be due to bacterial APA, which is more complex than for
phytoplankton, in that it can be linked to the hydrolysis and acquisition of C (Hoppe, 2003). This
is supported by the initial decrease in DOP and increase in $PO_4^{3-}$, which indicates that the AP
hydrolysis of DOP added to the $PO_4^{3-}$ pool. It is known that APA stays suspended and active for
a long time in marine environments, and cell-free APA was reduced by only 25% over 16 days in
the experiment by Thomson et al. (2019). If this enzyme is viable for this long, it suggests that
there was no new production of AP after the closure of the mesocosms, which is supported by
the dilution effect of adding the deep-water. In that case, the disappearance of the initial AP took
30 days.
The hydrolysis rates of AP were relatively low compared with most published data, probably
linked to the clear surplus of $PO_4^{3-}$. It is worth to note, however, that we were most likely not



measuring the maximal potential hydrolysis rates as substrate addition was relatively low (100
nmol L$^{-1}$) and would likely have been higher with more added substrate. This could be the reason
for the apparent discrepancy between the measured hydrolysis rates and the change in the PO$_4^{3-}$
and DOP pools during the 10 first days of the experiment. During this time there was a decrease
of approximately 0.5 µmol DOP L$^{-1}$ and an increase of 0.6 µmol PO$_4^{3-}$ L$^{-1}$, suggesting an actual
hydrolysis rate of 2.0-2.5 nmol L$^{-1}$ h$^{-1}$ (assuming 500-600 nmol over 10 days). This is a factor 3-
4 higher compared with the initially measured APA of ~0.6 nmol L$^{-1}$ h$^{-1}$.
The statistical model that we applied was better at explaining the APA variation than LAP. APA
gradually decreased during the initial phase of the experiment to undetectable levels after the
middle of the experiment. Any correlation does not mean causality and the higher coefficient of
determination is probably rather a reflection of the clear temporal development in APA. If the
AP was produced before the closure of the mesocosm and slowly degraded as discussed above,
any connection with the biogeochemical or plankton community must be due to unrelated
temporal development. For example, the DIN also decreased over time but was likely not related
to the APA.
For the LAP the overall explanatory power by the biogeochemical and plankton community
composition was less than for APA, but interestingly the bacterioplankton community
composition clearly explained the variability better (38%) than the combined biogeochemical
and phytoplankton community (28%). Considering that the bacterial community was not
sampled as frequently as the biogeochemical variables and flow cytometer counts, we suspect
that the explanatory power would have increased with more frequent sampling. It is likely that
bacteria were producing the LAP and some taxa are more reliant on enzyme production for
nutrient acquisition than others (Ramin and Allison, 2019). Some dinoflagellates are also known



to produce LAP and most of the mesocosms with high dinoflagellate biomass except M4.
However, the phytoplankton community only explained 18% of the variability in LAP, and these
dinoflagellates were likely not producing any substantial amount of this enzyme.
In conclusion, we found very high levels of LAP (mostly in the range 200 – 800 nmol L$^{-1}$ h$^{-1}$),
which is an order of magnitude higher than most literature data. This is probably linked to the
upwelling supporting high levels of microbial activity in combination with the general DIN
limitation in the coastal Peruvian upwelling. There was measurable APA at the start of the
experiment, but this gradually declined to undetectable levels in all of the mesocosms midway
(~30 days) in the experiment. With high concentrations of $PO_4^{3-}$, low APA is not surprising, and
AP is a relatively slowly degrading enzyme that could have been fully dissolved and produced
before the closure of the mesocosms. Our statistical mode explained more of the variability of
APA (74%) compared with LAP, probably due to its clear temporal development. The
bacterioplankton community composition explained best the variability of LAP (38%) compared
with the combined biochemical and phytoplankton community model (28%). With more than
50% of the variability unaccounted for, we are still missing important pieces of the puzzle
understanding the variability in this enzyme. The hydrolysis rates for LAP suggests that pelagic
N remineralization supported the relatively high standing stock of primary producers (mostly >4
µg chl-$a$ L$^{-1}$) in the mesocosms after N depletion.


Data availability



All data will be made available on the permanent repository www.pangaea.de after publication.
The DNA sequencing data will be submitted to NCBI SRA (in prep).

Author contribution
Samples were taken by KS, JP, JA, LB, EvdE, MF, NHH, JM and UR. In addition to the
sampling crew, further data analysis was conducted by MTCG and MGL. UR developed the
experimental design and sampling strategy and coordinated the mesocosm campaign. All co-
authors contributed to the data interpretation. KS wrote the manuscript with contributions from
all co-authors.

ACKNOWLEDGEMENTS
The experiment was funded through the German Research Foundation (DFG) project:
Collaborative Research Centre SFB 754 Climate1074 Biogeochemistry Interactions in the
Tropical Ocean. Additional funding came from the Academy of Finland (decision 259164; KS
and JP), the EU project AQUACOSM (UR) under grant No. 731065 of the European Union's
Horizon 2020 research and innovation programme, the Leibniz Award 2012 of the German
Research Foundation (UR) and the Helmholtz International Fellow Award 2015 (JA). This study
also used SYKE marine research infrastructure as a part of the Finnish FINMARI consortium.
The publication of sequencing data was approved by the Peruvian Ministry of Production with
respect to the access and benefit sharing regulations of the Nagoya protocol.





We thank all participants of the KOSMOS-Peru 2017 study for assisting in mesocosm sampling
and maintenance in particular: Andrea Ludwig, Jana Meyer, Jean-Pierre Bednar, Gabriela
Chavez, Susanne Feiersinger, Peter Fritsche, Paul Stange, Anna Schukat and Michael Krudewig.
We are particularly thankful to the staff of IMARPE for their support during the planning, and to
the Marina de Guerra del Perú, the Dirección General de Capitanías y Guardacostas and the Club
Náutico Del Centro Naval for their great support. The NMDS plots and GAM models were
produced by Dr. Riina Klais-Peets at EcoStat ltd.

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



Figure legends

Fig 1. The concentration of dissolved inorganic nitrogen (DIN), phosphate ($PO_4^{3-}$), dissolved
organic nitrogen (DON) and phosphorus (DOP). The red and blue color are the mesocosm bags
with deep-water addition with low (closer to shore) and very low (further offshore) oxygen
minimum zone (OMZ) signature, respectively. The green dashed lines denote the time of OMZ
water addition.

Fig 2. The fluorescence dissolved organic matter (FDOM) components (C1-C4) during the
experiment. The red and blue color are the mesocosm bags with deep-water addition with low
(closer to shore) and very low (further offshore) oxygen minimum zone (OMZ) signature,
respectively. The green dashed lines denote the time of OMZ water addition.

Fig 3. The Chlorophyll-*a* (Chl-*a*) concentration (upper graph) and the photochemical efficiency
(lower graph). The red and blue color are the mesocosm bags with deep-water addition with low
(closer to shore) and very low (further offshore) oxygen minimum zone (OMZ) signature,
respectively. The green dashed lines denote the time of OMZ water addition.




Fig 4. Development of the main groups of phytoplankton enumerated by flow cytometry. The red
and blue color are the mesocosm bags with deep-water addition with low (closer to shore) and
very low (further offshore) oxygen minimum zone (OMZ) signature, respectively. The green
dashed lines denote the time of OMZ water addition.


Fig 5. The bacterial community composition in the 8 mesocosms taken at different time points.
In the upper row are mesocosms with deep-water from low OMZ signature (30 m depth) and in
the second row with very low OMZ signature (90 m depth). The Y-axis indicates the relative
abundance of the bacterial taxa. Only the groups that contributed more than 0.5 % of the total
sequences are included and the rest are grouped as "Other Bacteria". The classification was
performed mainly in class, order and genus levels. The abbreviations indicate the main class
levels: Alphaproteobacteria (orange shades), Gammaproteobacteria (blue-pink shades),
Deltaproteobacteria (green shades), and Bacteroidia (yellow shades) .

Fig 6. The leucine aminopeptidase (LAP) and cumulative LAP activity. The red and blue color
are the mesocosm bags with deep-water addition with low (closer to shore) and very low (further
offshore) oxygen minimum zone (OMZ) signature, respectively. The green dashed lines denote
the time of OMZ water addition.




Fig 7. The alkaline phosphatase activity (APA) and cumulative APA. The red and blue color are
the mesocosm bags with deep-water addition with low (closer to shore) and very low (further
offshore) oxygen minimum zone (OMZ) signature, respectively. The green dashed lines denote
the time of OMZ water addition.


Fig 8. Non-parametric multidimensional scaling (NMDS) plots for biochemical, phytoplankton
community and bacterioplankton community (upper row). From the NMDS scores, generalized
additive models (GAMs) were made (lower two rows) where we used alkaline phosphatase
activity (APA) and leucine aminopeptidase (LAP) as dependent variables. The output scores
(mds1 and mds2) of the NMDS are depicted in the lower two rows.







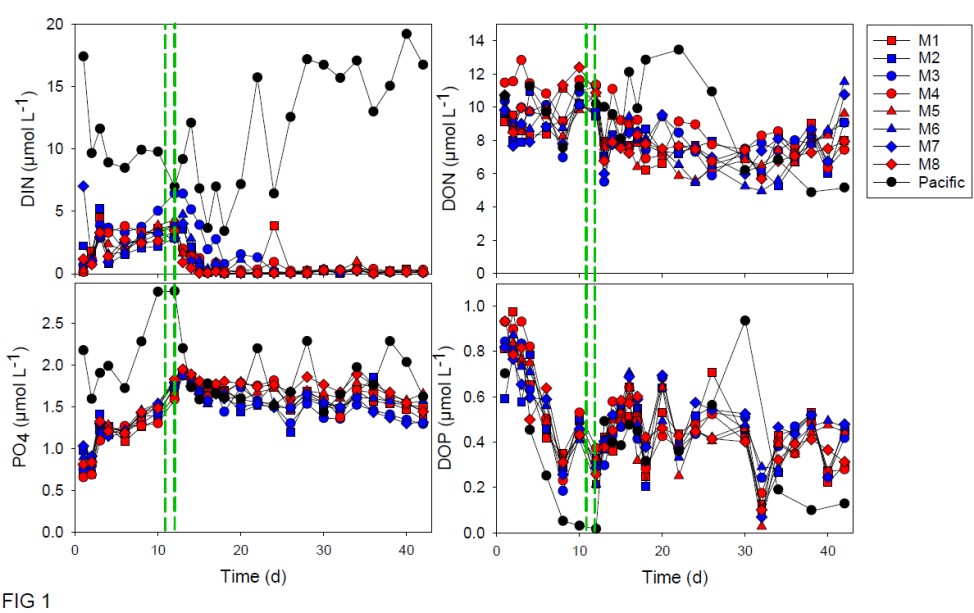

FIG 1




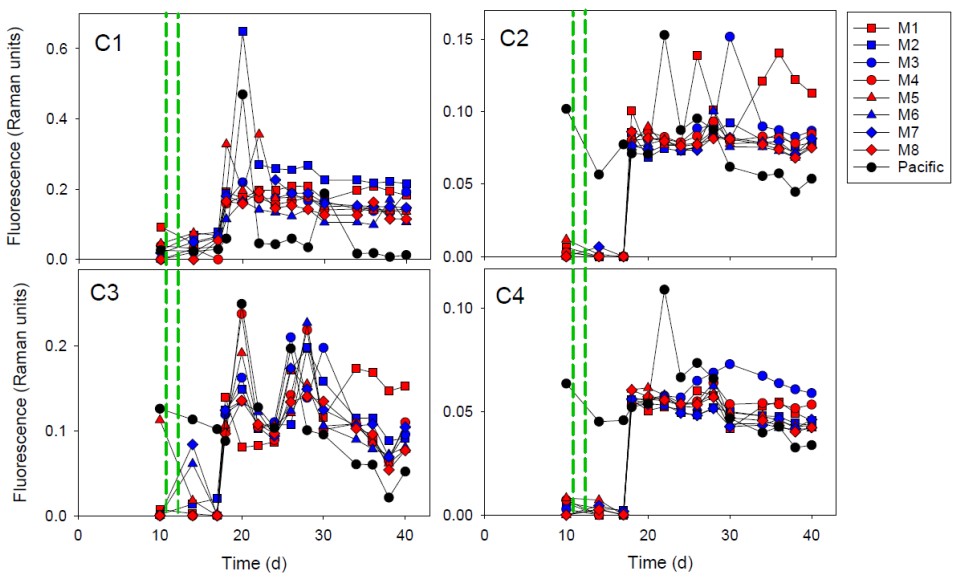

FIG 2





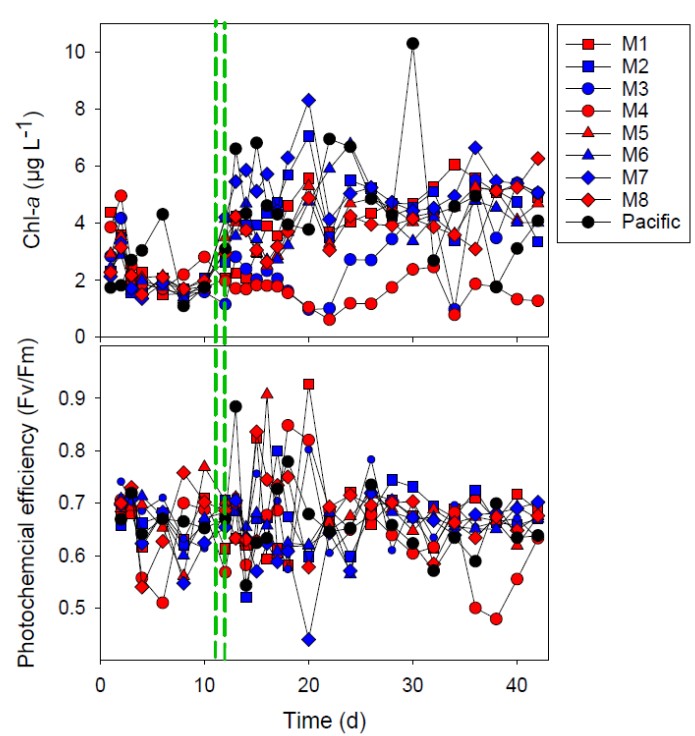

FIG 3




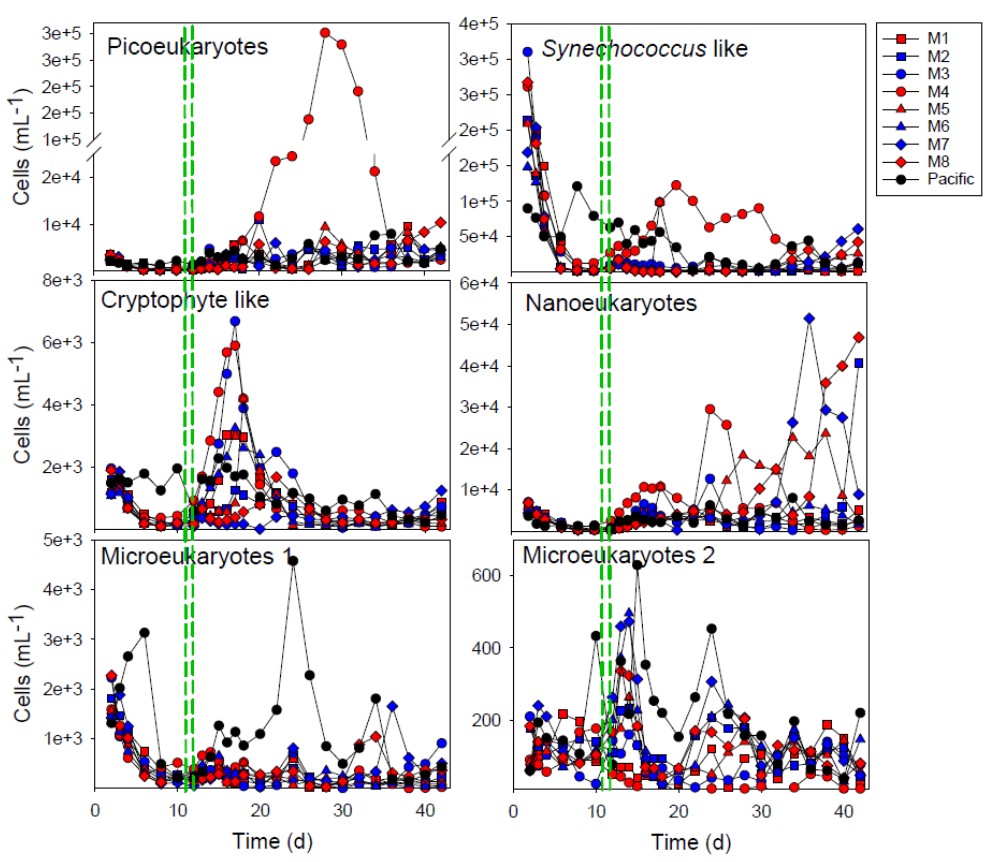

FIG 4






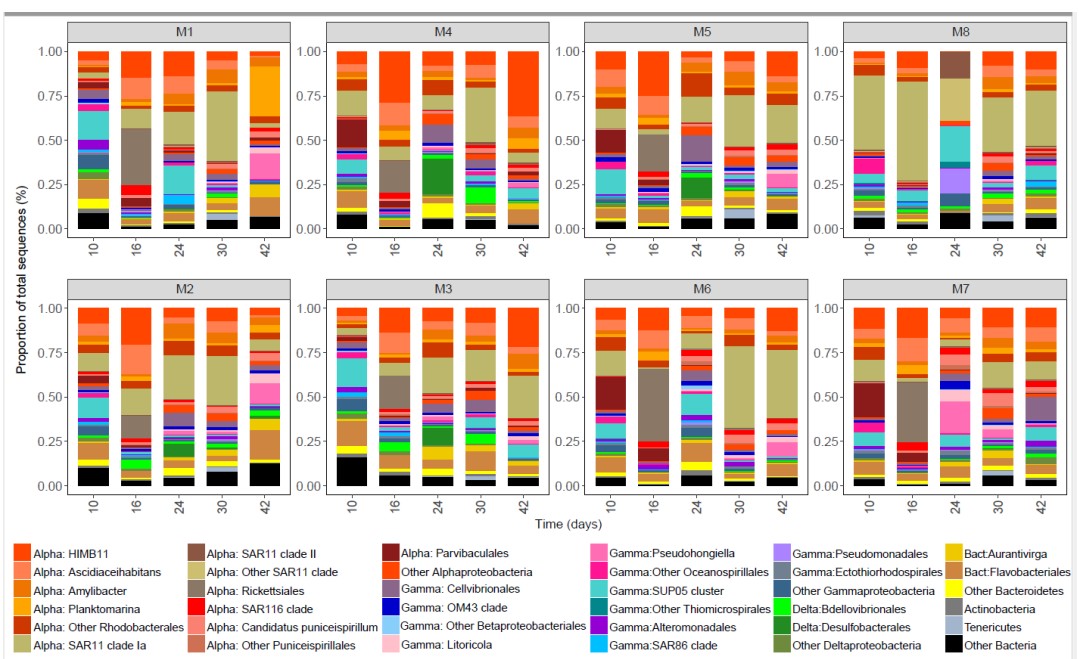


FIG 5





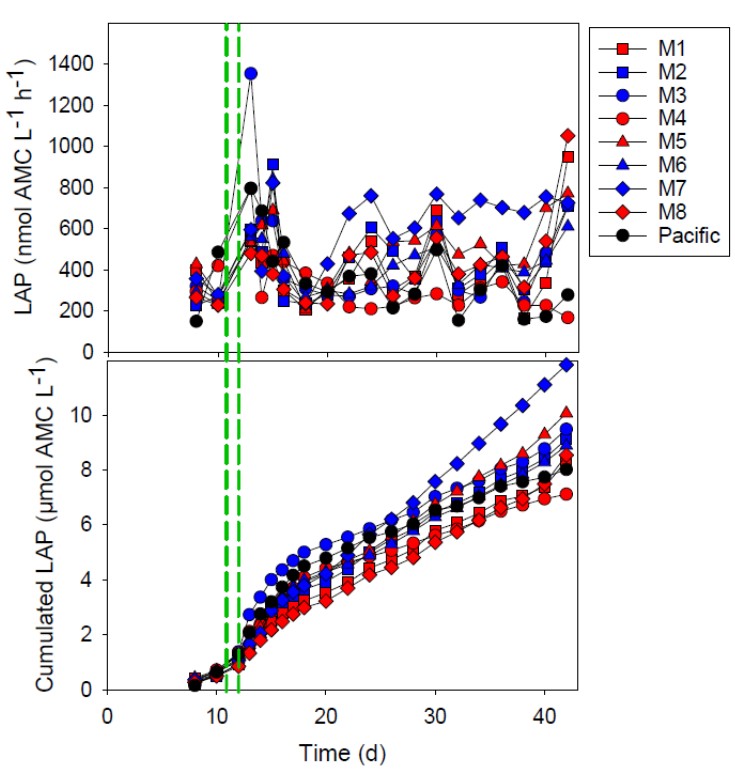


Fig 6



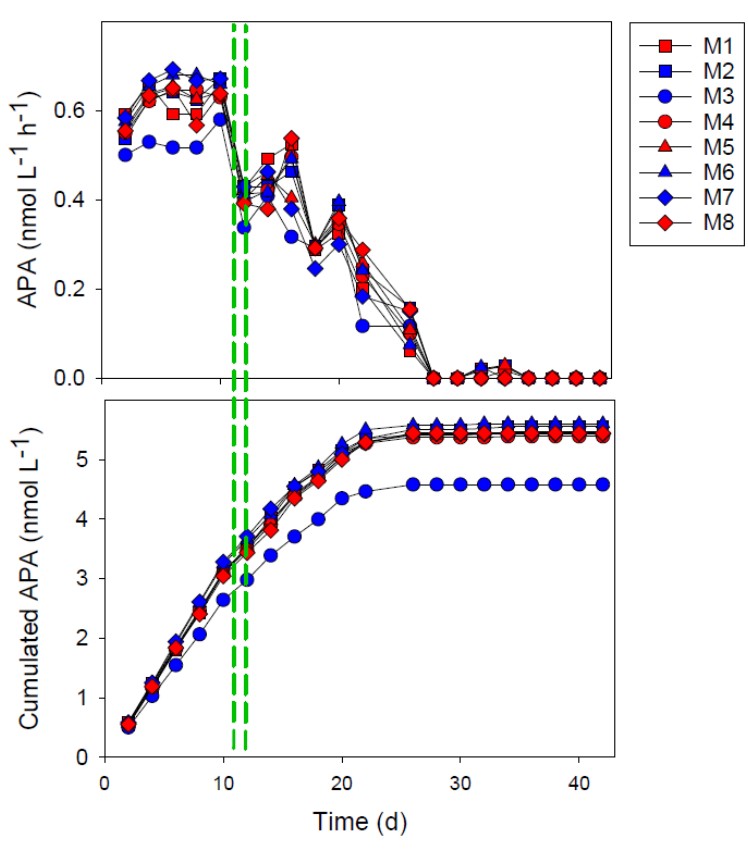


Fig 7



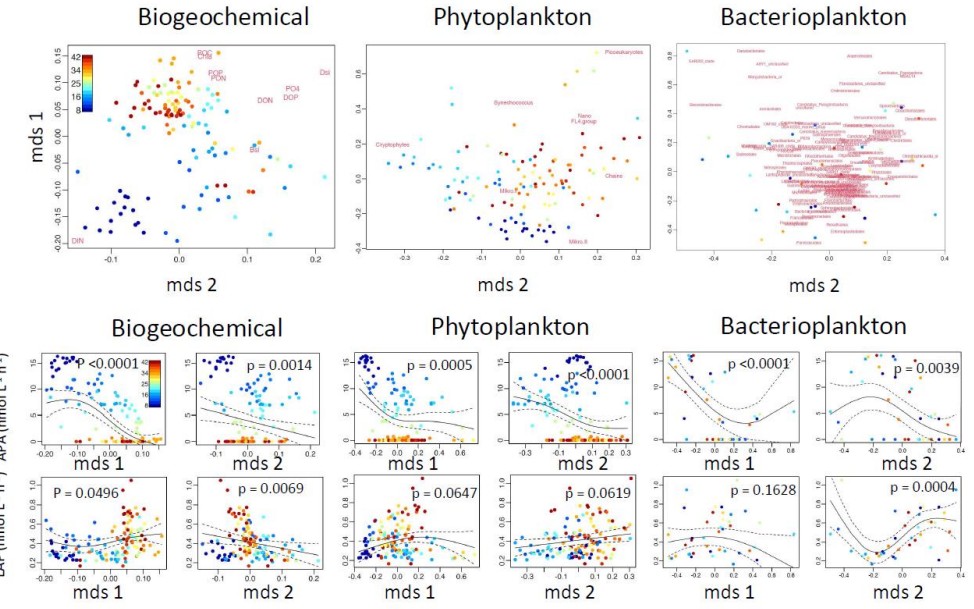


FIG 8