# Peer review of "Extracellular enzyme production in the coastal upwelling system off Peru during different upwelling scenarios: a mesocosm experiment"

_Biogeosciences, 2022_

## Referee Comment (RC1)

Comments to the manuscript submitted to "Biogeosciences"

entitled

"Extracellular enzyme production in the coastal upwelling system off Peru during different upwelling scenarios: a mesocosm experiment"

By Ch. Spilling et al.

After reading the manuscript, I have the following ideas about the experiments carried out: Originally, the influence of upwelling water on the biogeochemical processes in surface water was apparently to be investigated. However, due to the El-Nino event, no upwelling water was available. Therefore, water from the OMZ was used to simulate the effect of nutrient-rich water upwelling into the surface layer. The OMZ water was filled into the deeper layers of the mesocosms and was kept there during the duration of the experiment. The 0 – 10 m layer above was sampled. Consequently, from my point of view, the influence of the OMZ on the overlying water layer was investigated. This is a very interesting subject. To what extent effects of upwelling water can be inferred from the effects of OMZ water could be part of the discussion. Upwelling water reaches the water surface directly and mixes with surface water during further transport and does not layer underneath. However, the biogeochemical properties could be compared using date from the literature.

The results part, especially the first chapter "Nutrients" is difficult to read. As emphasized in the title, the focus is on two scenarios. The description of the individual mesocosms does not make the two scenarios clear. You have to look constantly at the figures in order to assign the respective mesocosm to the corresponding scenario. Additionally, constantly different points in time are described in the course of the experiment (sometimes you are at the OMZ water treatment, then at the time before and then again at the end). That's even more confusing. For each scenario, 4 parallel mesocosms were examined. My suggestion would be to describe the general trend of each scenario. Boxplot charts for each sampling time and each treatment would perhaps better illustrate the trends in the two scenarios than the line charts.

In the time course of the parameters in the scenarios, a fixed structure should be maintained - the course before the addition of OMZ water, then the course in scenarios 1 and 2. In the discussion, the influence of different variables on the enzyme activity in the surface water could then be considered the addition of OMZ water and the individual OMZ treatments. In my view, such a structure would make for an interesting manuscript.

Further Comments:

Title:

The word „ production"  should be replaced by the word "activity" (in the abstract too) , because that is what was measured. From my point of view, no upwelling scenarios were examined.

Introduction:

Page 3 line 53: "The ongoing warming…"

Comment: What is meant: Warming due to climate change or warming of upwelling water during transport. Please formulate exactly.

Page 3. Line 45-46: "The fate of biomass…"

Comment: I would move this sentence to the end of the chapter (line 52) and continue with lines 63ff. The chapter describing the OMZ (line 53-62) could be placed before the objectives.

Material and methods:

Page 5, line 93 and throughout the entire ms: Please replace deep-water by OMZ-water, because, in the ocean, water from 90 m is not deep water.

Page 5, line 99: "…. And sampled every second day….."
Comment: Please change into "….. and sampled every second day over a period of 50days…"

Page 5 line 103: " … on day 5…"
Comment: the words can be deleted, they are not necessary here.

Page 5, line 103: ".. from 90m depth…"

Comment: According to Bach et al. 2020 water was taken from 70 m depth.

Page 7 , line 136-142: "To measure  inorganic nutrients, total dissolved nitrogen (TDN) and phosphorus (TDP), the samples were first  filtered through pre-combusted (5 h, 450°C) Whatman GF/F filters (pore size 0.7 µm). The filtrate was collected in 50 mL acid-cleaned high-density polyethylene (HDPE) bottles and placed directly into a freezer (-20°C). Later the filtrates were thawed at room temperature over a period of 24 hours and divided in two. The first half was used to determine inorganic nutrient concentrations as described above. From the other half we determined the TDN and TDP concentrations. "

Comment: I would move this paragraph to the beginning of the Nutrient chapter

Page 9, line 192: Which extraction agent was used? Please provide a reference for the chlorophyll extraction and measurement

Page 12-13: Measurement of enzyme activity: how many parallels (pseudo-parallels) are measured for each sample?

Page 13, line 261-264: Microtiter plates were obviously used for the enzyme activity measurements, as has been described several times. I don't see how it can be done in 20ml Subsamples. Please correct the error.

Results:

According to Bach et al. In 2020 scallop larvae were added to the treatments. It should be mentioned whether the addition of these organisms affects the enzyme activities.

Discussion:

Page 23-24: line 499-502: " The hydrolysis rates of AP were relatively low compared with most published data, probably linked to the clear surplus of PO4 3-. It is worth to note, however, that we were most likely not measuring the maximal potential hydrolysis rates as substrate addition was relatively low (100 nmol L-1) and would likely have been higher with more added substrate.

Comment: For the LAP activity, it was described in the "Methods" that the substrate saturation concentration was determined before. I assume that was done for the AP activity as well. If so, then higher substrate loading should only marginally increase the AP activity.

In summary:

The authors should consider whether upwelling water effects were actually simulated or whether the manuscript should be geared towards OMZ effects. The manuscript should then be focused accordingly. The current manuscript is not well focused and sometimes difficult to read.

---

## Author Response (AR1)

**Response to the review**

We thank the reviewer for constructive comments and have addressed all the questions and comments below as author response (AR) in red.

Reviewer #1

After reading the manuscript, I have the following ideas about the experiments carried out: Originally, the influence of upwelling water on the biogeochemical processes in surface water was apparently to be investigated. However, due to the El-Nino event, no upwelling water was available. Therefore, water from the OMZ was used to simulate the effect of nutrient-rich water upwelling into the surface layer. The OMZ water was filled into the deeper layers of the mesocosms and was kept there during the duration of the experiment. The 0 – 10 m layer above was sampled. Consequently, from my point of view, the influence of the OMZ on the overlying water layer was investigated. This is a very interesting subject. To what extent effects of upwelling water can be inferred from the effects of OMZ water could be part of the discussion. Upwelling water reaches the water surface directly and mixes with surface water during further transport and does not layer underneath. However, the biogeochemical properties could be compared using date from the literature.

*AR: The original idea was to add collected and add deep water from two different locations with different degrees of OMZ signature. The difference between the OMZ water collected from the two sites was much smaller than anticipated. In this coastal area the OMZ is very close to the surface, and you are right we did add the OMZ water to the bottom of the mesocosm bags simulating this and following the exchange with the upper (0 – 10 m) part, and not direct upwelling as you point out.*

The results part, especially the first chapter "Nutrients" is difficult to read. As emphasized in the title, the focus is on two scenarios. The description of the individual mesocosms does not make the two scenarios clear. You have to look constantly at the figures in order to assign the respective mesocosm to the corresponding scenario. Additionally, constantly different points in time are described in the course of the experiment (sometimes you are at the OMZ water treatment, then at the time before and then again at the end). That's even more confusing. For each scenario, 4 parallel mesocosms were examined. My suggestion would be to describe the general trend of each scenario. Boxplot charts for each sampling time and each treatment would perhaps better illustrate the trends in the two scenarios than the line charts.

*AR: You are right, and we have focused more on the temporal development rather than difference between the two treatments (e.g. removed this from the title) and we have also rewritten the "Nutrients" paragraph to make it clearer.*

In the time course of the parameters in the scenarios, a fixed structure should be maintained - the course before the addition of OMZ water, then the course in scenarios 1 and 2. In the discussion, the influence of different variables on the enzyme activity in the surface water could then be considered the addition of OMZ water and the individual OMZ treatments. In my view, such a structure would make for an interesting manuscript.

*AR: Thank you for the comment. We have tried to change the focus slightly to be more about the temporal development rather than comparing treatments.*

Further Comments:

Title:

The word „ production"  should be replaced by the word "activity" (in the abstract too) , because that is what was measured. From my point of view, no upwelling scenarios were examined.

*AR: We changed production to activity*

Introduction:

Page 3 line 53: "The ongoing warming…"

Comment: What is meant: Warming due to climate change or warming of upwelling water during transport. Please formulate exactly.

*AR: Added:" …caused by climate change…"*

Page 3. Line 45-46: "The fate of biomass…"

Comment: I would move this sentence to the end of the chapter (line 52) and continue with lines 63ff. The chapter describing the OMZ (line 53-62) could be placed before the objectives.

*AR:  Changed according to suggestion*

Material and methods:

Page 5, line 93 and throughout the entire ms: Please replace deep-water by OMZ-water, because, in the ocean, water from 90 m is not deep water.

*AR: Good point, we did this change throughout the manuscript*

Page 5, line 99: "…. And sampled every second day….."

Comment: Please change into "….. and sampled every second day over a period of 50days…"

*AR: Added this to the sentence, that was moved and expanded based on comments from Rev#2*

Page 5 line 103: " … on day 5…"

Comment: the words can be deleted, they are not necessary here.

*AR: Deleted*

Page 5, line 103: ".. from 90m depth…"

Comment: According to Bach et al. 2020 water was taken from 70 m depth.

*AR: Correct, changed to 70 m depth*

Page 7 , line 136-142: "To measure  inorganic nutrients, total dissolved nitrogen (TDN) and phosphorus (TDP), the samples were first  filtered through pre-combusted (5 h, 450°C) Whatman GF/F filters (pore size 0.7 µm). The filtrate was collected in 50 mL acid-cleaned high-density polyethylene (HDPE) bottles and placed directly into a freezer (-20°C). Later the filtrates were thawed at room temperature over a period of 24 hours and divided in two. The first half was used to determine inorganic nutrient concentrations as described above. From the other half we determined the TDN and TDP concentrations. "

Comment: I would move this paragraph to the beginning of the Nutrient chapter

*AR: This paragraph only contains information on the total N and P plus DON and DOP so we think it is better suited here after the inorganic DIN and PO4.*

Page 9, line 192: Which extraction agent was used? Please provide a reference for the chlorophyll extraction and measurement

*AR: Acetone, and we added this information with a reference.*

Page 12-13: Measurement of enzyme activity: how many parallels (pseudo-parallels) are measured for each sample?

*AR: For LAP: Due to capacity limitations, primarily limited amount of substrate available, we did only one measurement per sample. For APA there was also only one measurement made per sample.*

Page 13, line 261-264: Microtiter plates were obviously used for the enzyme activity measurements, as has been described several times. I don't see how it can be done in 20ml Subsamples. Please correct the error.

*AR: The activity was added to 20 ml subsamples, that was further distributed into the wellplate for the measurements of fluorescence. We rewrote to make this clearer.*

Results:

According to Bach et al. In 2020 scallop larvae were added to the treatments. It should be mentioned whether the addition of these organisms affects the enzyme activities.

*AR: As was also mentioned in Bach et al 2020:*

*"However, few scallop larvae and no fish larvae were found in the mesocosms after the release so that their influence on the plankton community should have been small and will only be considered in specific zooplankton papers in this special issue."*

*We do not think it affected the results in any way and decided to omit this as to not confuse the reader.*

Discussion:

Page 23-24: line 499-502: " The hydrolysis rates of AP were relatively low compared with most published data, probably linked to the clear surplus of PO4 3-. It is worth to note, however, that we were most likely not measuring the maximal potential hydrolysis rates as substrate addition was relatively low (100 nmol L-1) and would likely have been higher with more added substrate.

Comment: For the LAP activity, it was described in the "Methods" that the substrate saturation concentration was determined before. I assume that was done for the AP

activity as well. If so, then higher substrate loading should only marginally increase the AP activity.

*AR: The approach taken in the LAP and APA were a bit different, with the LAP activity we wanted the maximum potential and did initial tests to find the LAP concentration needed. For APA this was done with a standard concentration used by the group doing these measurements and it was likely not providing the maximal potential hydrolysis rate.*

In summary:

The authors should consider whether upwelling water effects were actually simulated or whether the manuscript should be geared towards OMZ effects. The manuscript should then be focused accordingly. The current manuscript is not well focused and sometimes difficult to read.

*AR: We agree it is a bit 'messy' dataset with a lot of different angles. The original goal of the experiment was to study the effect of OMZ water with different signatures and we argue for keeping that. However, we have changed the text a bit to better reflect the temporal development and the effect over time on the enzyme activity.*

Reviewer #2

Spilling et al. report a series of measurements from highly complex mesocosm experiments, in which water was initially filled into multiple mesocosms, supplemented part way through with other water, and followed for a considerable time period. Doing fieldwork – especially with mesocosms – is a difficult and often frustrating task, since there is inherent variability in natural waters, and often one (or more) mesocosms will go their own way, despite scientists' efforts to have multiple replicates of specific treatments. From this perspective, it is understandable that Spilling and colleagues have a somewhat messy data set in which multiple parameters were measured from many mesocosms throughout a time series; figuring out what story the data are telling one is not an easy task.

However, the authors really need to spend a bit more time with their data in order to understand how the pieces do – and don't – fit together, and above all, in order to make the readers' journey through the manuscript as straightforward as possible. In the current version, essential information (e.g., depths from which the water for various analyses was collected) is missing or hard to find, and after reading the Methods and the Results, the reader is left confused, instead of having a general roadmap as to which parameters are being compared and how the treatments may or may not make a difference. For example, from just looking at the figures, it appears that the red and blue mesocosms (low and very low OMZ water addition) do not differ from one another in a systematic manner – sometimes the M4 mesocosm is an outlier, but not always; sometimes the M1 mesocosm might be an

outlier. Does the main story lie then in the time course of evolution of these different mesocosms? Reading Bach et al leaves the impression that there might be a story in this direction; certainly the differences between the low and very low OMZ water addition does not seem to show robust differences.

*AR: The reviewer points to some of the same things as reviewer #1. I think it is fair to say that the experiment did not go according to the original plan as we did not get the different properties in the two OMZ water that was collected and put into the mesocosms. That said, we argue that the data present valuable information on the planktonic ecosystem in the Peruvian upwelling that is important globally in terms of fish catch.*

*We rewrote parts of the materials and methods. The information was mostly there, but perhaps not clearly enough pointed out, for example that all the samples were taken with an integrated water sampler, collecting the samples from 0 to 10 m depth.*

*There is not much difference between treatments, and we have changed the manuscript to convey the more time series development of slight differences in the OMZ signature water that was added.*

Specific comments:

The abstract does not follow a clear line – it reads like a listing of observations/parameters. The authors need to portray a more coherent overview. As an example, the first sentence of the abstract discusses climate change – if this point is not carried through the manuscript, alter the introduction of the abstract so that it points the reader in the direction that does.

*AR: We rewrote the abstract and removed the reference to the climate change.*

The explanation of the setup of the mesocosms and introduction of deep water was extremely confusing. Why was water exchanged in the mesocosms?

*AR: The bags can only contain a specific volume of water. Once filled, some of the water had to be removed before the OMZ water could be added. We tried to make this point clearer.*

The authors should also discuss the effects of the introduction of brine on the microbial community. From reading Bach et al 2020, it seemed to be a very strong brine solution, so the activity/composition of organisms in this lower part of the water column was probably considerably affected. Much of the brine addition part of the manuscript (lines 117-124) was only really understandable after reading Bach

et al 202; this part of the manuscript should be re-worked so that the main points are clear without reading the other manuscript for details.

*AR: The brine was quickly diluted, and the salinity was mostly less than 1 PSU different below the halocline that was created. We did only sample above this halocline, which is better explained now. The potential effect of the brine addition is likely very small, but there was a slow increase in salinity also in the upper sampled part (0-10 m) of the mesocosm bags and we now mention this in the discussion.*

Note also that any data that are re-used from Bach et al 2020 (at least some of the nutrient data?) should be noted in the methods.

*AR: True, some of the background data like nutrients were also presented in Bach et al 2020 and we have now mentioned this in the methods section.*

It is very difficult to figure out which part of the mesocosm was measured – where did the water come from (which depths) for each analysis? This information is unclear for measurements of nutrients, FDOM, flow cytometry, chl a, and L-MCA measurements. Only the sequencing description also explicitly includes this information (line 199).

*AR: This was mentioned in L 99-100, but we agree that it could be better explained. We now moved this into a separate paragraph to really highlight how this was done.*

*"Sampling took place every second day over a period of 50 days, and all variables, unless stated otherwise, were taken with an integrated water sampler (HydroBios, IWS) pre-programed to fill from 0 – 10 m depth. These integrated water samples (0 – 10 m) were stored dark in cool boxes and brought back to the laboratory and processed right away. Sampling took place in the morning and the samples were usually back in the lab around noon. "*

Detailed comments

Line 23: "....extracellular enzyme production of leucine aminopeptidase..." (measured activities, not production of the enzyme; reword)

*AR: Corrected*

Line 30: note that LAP does not degrade amino acids; it hydrolyzes terminal amino acids from larger units (the N-terminus of peptides or proteins). The amino acids themselves are degraded by other enzymes.

*AR: Thanks for noting, we corrected this.*

Line 64: organisms are productive, surface layers are not. Reword "the productive surface layer is driven by recycled production."

*AR: True, we changed this to : ...primary production in the surface layer...*

Line 71: wording such as "two of the most studied ones" requires references as examples, for example ' (e.g. author 1, year; authors et al. , year)'

*AR: We added some references*

Line 73: note that the Leu-MCA substrate integrates the activities of a wide range of peptidases (Steen et al. 2015, Substrate specificity of aquatic extracellular peptidases assessed by competitive inhibition assays using synthetic substrates Aquatic Microb Ecol 75:271. In any case, there are also a wide range of leucine aminopeptidase enzymes, so it is not 'a' protein degrading enzyme.

*AR: Yes, we changed this and included the suggested reference also.*

Lines 108-116: explain explicitly why the deep water was put in on days 11. The entire water exchange section is very confusing; rewrite or put in a figure as a supplement to guide the reader as to which water had what characteristics

*AR: The whole experiment was set up to simulate different upwelling scenarios and the deep water was added for this purpose. It did not go completely according to plan as the collected OMZ water from the two different locations was more similar than expected due to the coastal El Niño event. We did rewrite this section to make it clearer.*

Line 204: wording: what is meant by 'properly' homogenizing a sample?

*AR: Excess wording, we removed the word 'properly'.*

Line 252: How much seawater was added to each replicate, compared to the L-AMC solution? How many replicates were measured? What was the maximum time of incubation (only minimum is given)? Was fluorescence of killed controls subtracted from the live incubations?

*AR: 200 µl (180 µl sample + 20µl substrate) same as for the L-AMC solution. One measurement per sample. The incubation period was four to six hours. It was not possible to use e.g. formalin to kill samples in the location we worked, but we did blank subtraction of the sample without substrate addition. We added all this information to the methods section.*

From what depth was the water used to measure L-AMC activity? How much time elapsed between sample collection and measurements of enzyme activities?

*AR: We have added this information to the general sampling description. All samples were from the integrated 0- 10 m sample. Samples were taken in the morning and started to be processed after returning to the lab around noon.*

Line 261: Where was the 20 ml subsamples obtained? What depth was this water collected at?

*AR: See previous comment*

Line 273: what are the two sample treatments (at this point in the manuscript, the differences are not clear)

*AR: The addition of two different OMZ water (from two different locations). We added this information to make it clear.*

Line 296: It is not clear from which depths the measurements were made, so the effects of the addition of the deep water are a bit confusing.

*AR: This is now better explained in the materials and methods, it originates from the integrated 0 – 10 m depth.*

Line 326: at what depth was chl a measured?

*AR: Same as for the previous comment*

Line 343: Are any sequences available for the initial mesocosm, or for water outside the mesocosms? Why were these particular time points selected for sequencing?

*AR: There was no way to check the sequences during the experiment so the time points had to be pre-set. Due to the cost of sequencing it was not possible to do the full set so we had to limit the number of samples taken. This is also the reason for not including the water from outside the bags.*

Line 366: What is the difference between deep water and OMZ water? The varying terminology is confusing.

*AR: Yes, this was something raised by Reviewer #1 as well and we now changed this OMZ water only.*

Note that a rate of 359 nmol L-1 h-1 is not low – it is far higher than most rates reported in the literature for water column measurements of LAP.

*AR: Yes, but we write relatively low, meaning compared with the other LAP activity data we present in the paper.*

Line 369: what is the rationale for plotting 'cumulative LAP' activities? Presumably if rates had been measured at even more time points, then the cumulative LAP rates would have been even higher, but it is difficult to understand the biological or biochemical rationale for summing the rates in this fashion.

*AR: The main rational for using cumulative enzyme activity is to see differences between mesocosms more clearly, especially if there is a lot of variability between sampling days. For example, in Fig 6 it is easy to spot M6 as a mesocosm with more LAP in the second half of the experiment. It was also used to compare treatments statistically by comparing the slope of the cumulative activity instead of a repeated measures approach.*

*In this case, it is perhaps not that much extra information gained by including the cumulative value, so we decided to remove this panel from the figure and reference to the cumulative value in the text.*

Line 382 – line 390: use of statistics in this manner leaves the reader with the impression that the authors have run out of ideas. For example, the LAP activities are extraordinarily high. The statistical link with the bacterial community and biogeochemical variables is ok, but what underlying biological explanation would the authors like to put forward? This point would be far more interesting than just the statistics.

*AR: The challenge with the data set is that we have measured bulk values. This means that we are not able to pinpoint what organisms were responsible for producing the extracellular enzyme avidity. Using this statistical approach it is possible to test correlations between environmental and biological variables with the enzyme activity. This is also in the results section, and we do take up the issue of the high LAP activity in the discussion.*

Line 405: the authors state they wish to "relate the biogeochemical and microbial community to the extracellular enzyme activity and a more detailed description of the temporal development and biomass comparison of microbial groups will be presented elsewhere in this special issue (e.g. Bach et al., 2020)". Bach et al discuss the phytoplankton, but the bacterial data could use some discussion, since presumably they are also the source of the LAP enzymes.

*AR: Yes, good point. We do discuss the bacterial community composition and its development further down L418-449, but we now included more information on the potential for bacterial LAP activity.*

*"It is likely that bacteria were producing the LAP activity…"*

Line 449: first mention of integrated 0-10m sampling? Should have been easy to find in the Methods.

*AR: It was (L99-100) but perhaps not well enough pointed out: "...sampled every second day with integrated water samplers (0-10 m depth, IWS, Hydro-Bios).*

*We reformulated this in the Materials and Methods chapter as described above.*

Line 472: what was the incubation temperature of the LAP samples? How long was the interval between water sampling and measurement?

*AR: The temperature was 20-22 °C most of the time and we added this information to the line.*

The final part of the discussion contains considerable repetition.

*AR: This is the conclusion paragraph, and we argue for keeping a short summary here to help the reader.*

Fig 1 caption: what is the difference between deep water and OMZ water? In addition, should note that the Pacific water was measured from water outside the mesocosms (as explained in Bach et al, but this term should be explained here, the reader shouldn't have to continually refer to Bach et al.)

*AR: It was just referring to addition of water that had been collected below the surface. We are aware that 30 m and 90 m depth cannot be considered very deep, so we removed the term 'deep-water' throughout the text and now refer to OMZ water. We added the information about the Pacific water.*

Fig 2 caption: see above with respect to Pacific water measurements

*AR: Corrected, see previous comment*

Fig 5 is extremely hard to read. Maybe try a bubble plot, or group the colors in a non-random manner to make it easier to determine which sequences are which colors (use perhaps patterns on the colors to distinguish them)

*AR: These types of plots are always difficult to decipher, but the trick is that the order of the stacks is the same as the legend. We will also publish all of the data behind the graph in order to make it accessible for anyone who wants to have a better understanding of the details of the graph.*

Figs 6 and 7: cumulative enzyme activities are meaningless (they depend on the frequency of measurement), so these panels should be deleted.

*AR: Yes, if only using the measured data, but assuming the enzyme activity is the same per hour from one measurement to the next, or making a simple linear regression model, make it possible to estimate the total cumulative enzyme activity.*

*See also our comment above on this. We agree that it does not provide very much added value in this case, and we did remove these panels.*

---

## Author Response (AR2)

**Response to the review**

We thank the reviewer for constructive comments and noting the shortcomings that still were present in the manuscript. We have addressed all the issues below as author response (AR) in red.

Reviewer #1

The manuscript is now more clearly focused on the influence of OMZ water on the surface layer than it was previously in the manuscript. This is especially the case in the introduction and the description of the methods which has been expanded. This makes the manuscript more consistent and easier to understand. The illustrations describing the courses in the individual mesocosms have been retained. This was possibly done to be consistent with the other publications in this special volume. In my view, the results would be better expressed by summarising the 4 parallels of a treatment. Perhaps the authors could insert a sentence at the beginning of the results saying that the parallels of a treatment do not always develop in the same way in all parameters. Then this form of presentation would be more justified. It is already mentioned in the manuscript elsewhere, but should be placed here.

AR. Thank you, and good point. We added a sentence to the results chapter under the community paragraph:

"The parallels of the same treatment did not develop in the same way in all the mesocosms, and this was particularly evident from the phytoplankton community composition (Fig 4)."

I still recommend the following minor changes:
Page 5, line 89-91: „In this study, a mesocosm experiment off the coast of Peru was carried out to study the effect of upwelling of OMZ water to the surface, with several papers covering different aspects in this special issue.
Comment: The words "of upwelling" should be deleted

AR: deleted as suggested

Page 6, line 107: "The main aim of the experiment was to simulate different upwelling events."
Comments: In the authors' response to the reviewer's comments, the objective stated that effects of OMZ water on surface water should be studied.
The sentence should be changed because it is not consistent with the content of the manuscript.

AR: we agree and deleted the sentence

Page 15, line 319- 321: "The addition of OMZ-water increased the phosphate concentrations whereas the dissolved inorganic nitrogen (DIN) was >2 μmol L-1 in the mesocosms until after the addition of OMZ-water (days 11 and 12 of the experiment)". Comment: the word "after" should be deleted.

AR: deleted

Page 21, line 434-436: "This was also seen in our mesocosm as the dinoflagellate Akashiwo sanguinea, a mixotrophic species that may form red tides (Jeong et al., 2005; Badylak et al., 2014), that quickly appeared in most mesocosm after OMZ water was added with some exceptions."
Comment: With a small change, the sentence would sound better: "This was also seen in our mesocosm after OMZ water addition as the dinoflagellate Akashiwo sanguinea, a mixotrophic species that may form red tides (Jeong et al., 2005; Badylak et al., 2014), quickly appeared with some exceptions."

AR: changed according to suggestion

Page 24, line 514- 515:" In our experiment, the initial decrease in DOP and increase in PO4 3- indicates that the AP hydrolysis of DOP added to the PO4 3-pool."
Comment: The sentence should be changed marginally: In our experiment, the initial decrease in DOP and increase in PO4 3- concentrations indicates that P released by AP hydrolysis was added to the PO4 3-pool.

AR: Changed according to suggestion

When the suggested minor changes have been taken into account, the manuscript can be published.

Reviewer #2

Overall, the authors have done a very good job clarifying the experimental setup and focusing the manuscript.

For Fig. 6, it was difficult to tell which was the revised figure, presumably because the track changes pdf version had the old as well as new figures. In any case, three individual figures were shown for Fig. 6. Presumably the new figure that will be used is the one in the middle – the top figure had blocks of red and blue that obscured the actual data points, the bottom figure was the 'cumulative hydrolysis' that didn't make much biological sense.

AR: yes, this must be due to the track changes.

In terms of the text, some minor points of word choice and phrasing could be fixed:

Line 136: 'were' instead of 'where'

AR: corrected

Line 143: 'was' instead of 'were'

AR: corrected

Final word of Line 455: change 'that' to 'which'
AR: Corrected

Line 475: 'linked' to the nutrient availability? ('liked' is what is written, but it seems out of place)

AR: yes, corrected

Line 490: change to 'It is worth noting…'

AR: corrected
Line 498: change 'were' to 'was'
AR: corrected

Line 525: change to 'It is worth noting…'
AR: corrected

Line 559: change 'were likely' to 'was likely'
AR: corrected